

# Do Centres of Endemism provide a spatial context for predicting and preserving plant phylogeographic patterns in the Cape Floristic Region, South Africa?

Nicholas C. Galuszynski and Alastair J. Potts

Botany, Nelson Mandela University, Port Elizabeth, Eastern Cape, South Africa

Corresponding author
Nicholas C. Galuszynski,
nicholas.galuszynski@gmail.com

## ABSTRACT

**Aim:** The evolutionary forces that gave rise to the exceptional plant species richness of the Cape Floristic Region (CFR) have also likely played a role at the intraspecific level (i.e. plant populations)—and thereby generating shared phylogeographic patterns among taxa. Here we test whether plant populations in the CFR exhibit phylogeographic breaks across the boundaries between Centres of Endemism (CoEs). The boundaries between CoEs (derived from the distribution ranges of endemic taxa and currently mapped at a coarse, Quarter Degree Square scale) represent a spatial proxy for the evolutionary diversifying drivers acting on plant taxa in the CFR.
**Location:** The CFR, located along the southern Cape of South Africa.
**Methods:** Published phylogeographic literature were compiled and spatial patterns of genetic divergence re-analysed to assess the frequency at which CFR plant taxa exhibit phylogeographic breaks either (1) across or (2) within CoE boundaries. Population pairs from each study were compared across and within CoEs and scored as either exhibiting a phylogeographic break or not.
**Results:** Phylogeographic breaks in Cape plants were found to occur across the boundaries of CoEs more often than not. Significantly more population pairs exhibited phylogeographic breaks across CoE boundaries (506 of the 540, $\chi^2 = 886$, $p < 0.001$) and fewer breaks within CoEs (94 of 619, $\chi^2 = 300$, $p < 0.001$) than would be expected if there was equal probability of a genetic break occurring across CoE boundaries.
**Main conclusions:** The evolutionary forces that have produced and maintained the exceptional plant diversity in the CFR appear to have operated at the population level, producing similar patterns of phylogeographic structuring of plant lineages regardless of life history or taxonomy. This tendency for Cape plants to exhibit shared patterns of spatially structured genetic diversity that match the distribution of endemic taxa may assist CFR phylogeographers to streamline sampling efforts and test novel hypotheses pertaining to the distribution of genetic diversity among CFR plant taxa. Additionally, the resolution at which CoEs are mapped should be refined, which may provide a valuable tool for future conservation planning and the development of precautionary guidelines for the translocation of genetic material during species reintroductions and commercial cultivation of Cape endemic crops. Thus, to answer the question 'Do Centres of Endemism provide a spatial context for predicting and preserving plant phylogeographic patterns in the Cape Floristic

Region, South Africa?'—yes, CoEs do appear to be an important tool for Cape phylogeographers. However, the data is limited and more plant phylogeography work is needed in the CFR.

## INTRODUCTION

Understanding spatial patterns of biodiversity is necessary to protect ecosystems, biotic communities, and species of high conservation importance. Unfortunately, the underlying patterns of intraspecific genetic diversity are rarely given equal attention (*Coates, Byrne & Moritz, 2018*). Relying on species, in some cases sub-species, as the primary unit for conservation efforts, assumes species are largely homogeneous entities. However, species represent a continuum of adaptive and neutral processes operating across populations (*Stapledon, 1928*; *Turesson, 2010*; *Vance & Kucera, 1960*). Phylogeographic studies of Cape plant taxa have revealed the tendency for populations to exhibit genetic structuring over relatively short geographic distances (*Britton, Hedderson & Verboom, 2014*; *Caujapé-Castells et al., 2002*; *Galuszynski & Potts, 2020a*; *Lexer et al., 2014*; *Malgas et al., 2010*; *Pirie et al., 2017*; *Potts et al., 2013*; *Prunier et al., 2017*), suggesting that steep ecological gradients are sufficient barriers to dispersal in the Cape Floristic Region (CFR) to drive genetic differentiation amongst populations. However, intraspecific genetic variation is challenging to integrate into conservation planning in the CFR due to the expense, expertise and time involved in generating such data.

The CFR, located on the southern tip of southern Africa, is a winter rainfall region with exceptional plant species richness and endemism (*Goldblatt & Manning, 2002*). Many of the endemic species have particularly small ranges, making them especially susceptible to extinction (*Helme & Trinder-Smith, 2006*; *McDonald & Cowling, 1995*; *Rebelo et al., 2011*; *Trinder-Smith, Cowling & Linder, 1996*). Focussing on preserving the processes that maintain these local endemics, systematic conservation planning has identified the need to protect the large scale processes that have shaped the evolutionary history of the regions biota (*Cowling et al., 2003*; *Pressey, Cowling & Rouget, 2003*; *Rouget et al., 2003*). However, with limited knowledge of the spatial patterns and extent of intraspecific genetic variation within species of an already diverse flora, this level of biodiversity remains largely under-represented in CFR conservation strategies. This issue comes to the fore when dealing with processes that involve redistributing genetic material for commercial production or rehabilitation efforts—potentially compromising the genetic integrity of local populations (*Hufford & Mazer, 2003*; *Laikre et al., 2010*; *Potts, 2017*).

Recent studies describing phylogeographic structuring of CFR plants highlight the importance of integrating intraspecific genetic variation into the conservation planning of the region. Work on core Cape clades (sensu *Linder, 2003*) has consistently detected phylogenetic structuring, with intraspecific divergence occurring over small spatial scales:

*Erica* (*Segarra-Moragues & Ojeda, 2010*; *Ojeda et al., 2015*; *Van der Niet et al., 2013*), *Protea* (*Prunier et al., 2017*; *Prunier & Holsinger, 2010*), *Restio* (*Lexer et al., 2014*), and *Tetraria* (*Britton, Hedderson & Verboom, 2014*). Furthermore, in the cases of the widespread *Protea repens* (L.) L. (Proteaceae) (*Prunier et al., 2017*) and *Restio capensis* (L.) H.P. Linder and C.R. Hardy (Restionaceae) (*Lexer et al., 2014*), environmental shifts appear to be more important for isolating populations than geographic distance. These studies also commented on the transition between phytogeographic zones (*Goldblatt & Manning, 2002*) as an important predictor of environmental transitions—highlighting the potentially important linkages between phytogeography and phylogeography.

Assuming that a regional biota has (largely) experienced the same broad evolutionary pressures (geological and climate variability), patterns of phylogeographic structuring are likely to be shared among species (however, co-occurring species can exhibit discordant phylogeographic patterns, *Soltis et al., 2006*). Applying multi-species data to identify regional patterns of genetic divergence has proven particularly valuable in exploring the role of Pleistocene climate oscillations in shaping species distribution patterns elsewhere in the world (*Byrne, 2008*; *Hewitt, 2008*; *Sork et al., 2016*; *Turchetto-Zolet et al., 2012*) and has facilitated conservation planning for biodiverse floristic regions (*Byrne, 2007*). Unfortunately, phylogeographic studies are numerically and spatially limited in the CFR, often focusing on a small spatial extent as CFR taxa tend to have restricted geographic ranges. Thus, there is often little overlap in sampling domain among studies, hindering any attempts to conduct comparative phylogeography in the CFR. However, the diversity of the system has attracted extensive taxonomic work (*Treurnicht et al., 2017*), producing remarkable species distribution records. We test whether phytogeographic boundaries can be used to predict the positions of phylogeographic breaks.

Here we used the core CFR Centres of Endemism (CoEs) of *Bradshaw, Colville & Linder (2015)*, coupled with published phylogeographic studies, to test how often plant populations exhibit phylogeographic breaks across the boundaries between these CoEs. While alternative phytogeographic zones are available (*Goldblatt & Manning, 2002*; *Weimarck, 1941*), they lack objective and reproducible methods (*Bradshaw, Colville & Linder, 2015*). In contrast, the CoEs were identified using numerical methods, based on the co-distribution of CFR endemic taxa. Briefly, the approach used in *Bradshaw, Colville & Linder (2015)* comprised the following: the distributions of CFR endemic taxa, mapped at Quarter Degree Squares (QDS), were evaluated and all widespread species and/or species whose distribution were limited to single QDS were downweighted; and, taxon similarity among cells was then measured and clustered to produce spatially defined endemic communities, CoEs. While the CoEs share spatial congruence with the earlier (subjective) maps (Fig. 1), CoEs appear to have a finer resolution with more centres and sub-centres. More importantly, much of the increased resolution of the CoEs is concentrated in the western CFR, which consists of a mosaic of multiple small CoEs while the eastern CFR is dominated by two large CoEs (CoEs 4 (blue) and 5 (green) in Fig. 1).

This pattern of decreased size and increased abundance of CoEs in the western CFR reflects the distribution of species richness in the Cape. The number of localised endemic taxa and species diversity is greatest in the west and gradually decreases as one moves east;

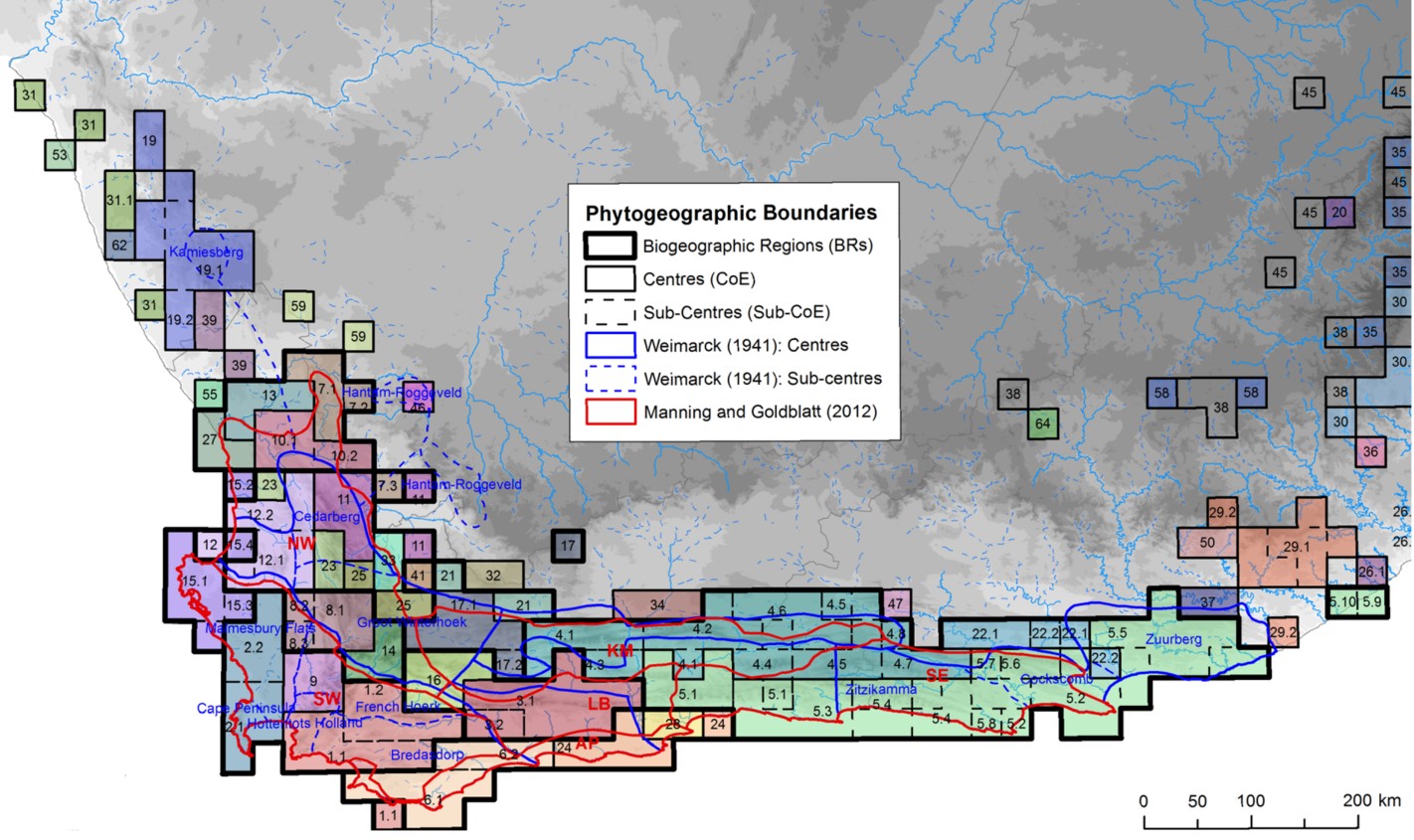

**Figure 1 Distribution of Centres and sub-Centres of Endemism of *Bradshaw, Colville & Linder (2015)* in relation to earlier phytogeographic zones (references given in image).** Phytogeographic zones are denoted by different line weights and colours following the description in the figure key. Each CoE, representing the co-distribution of CFR endemic plant species at a QDS resolution, is colourised and numbered on the map. Image source: *Bradshaw, Colville & Linder (2015)*.

this trend is referred to as Levyn's Law (after the renowned Cape botanist who first described this pattern, Margeret Levyn *Cowling et al., 2017*). This gradient of floral diversity has been ascribed to differences in climate history between the western and eastern CFR. The stable climate history of the western CFR would have promoted species accumulation through reduced extinction events (*Cowling, Proches & Partridge, 2009*; *Cowling et al., 2017*; *Cowling & Lombard, 2002*), with speciation occurring over short distances along niche axes (*Ellis et al., 2014*). In the eastern CFR, however, Pleistocene climate change disrupted vegetation distributions (*Chase & Meadows, 2007*; *Huntley et al., 2016*) resulting in possible extinction of local endemics (*Cowling & Lombard, 2002*), homogenising the Cape floral communities in this sub-region, and thus giving rise to fewer, large CoEs.

The overall stable evolutionary context in the CFR has possibly limited shifts in species ranges, thereby preventing gene flow among populations and promoting speciation (*Cowling et al., 2015*). As this context has not changed, populations are likely to become isolated over short distances and intraspecific genetic divergence may follow the distribution of endemic taxa. We therefore hypothesise that, intraspecific genetic divergence will mirror the distribution of CFR endemic taxa (i.e. phylogeographic breaks occur across the boundaries between CoEs more often than within CoEs), which represent

the ultimate consequence of genetic isolation over a short spatial scale: speciation. Ideally, sample distributions from a large number of studies with high density population sampling would be explored in the context of CoEs mapped at a scale finer than QDS, but such data is currently not available for the CFR. Nevertheless, by developing a simple rule set for assigning populations to CoEs and relying on published molecular analyses to detect phylogeographic breaks between population pairs, we tested the extent to which phylogeographic breaks mirror CoE boundaries in the CFR.

The insights presented here should facilitate future investigations into intraspecific genetic variation, wild genetic resource management, and conservation planning in the CFR (*Fraser & Bernatchez, 2001*; *Thakur, Schättin & McShea, 2018*). Furthermore, this study hopes to encourage Cape phylogeographers to design studies that test hypotheses regarding the spatial patterns of genetic diversity and the processes that maintain the exceptional diversity of this region and, aid in setting a precautionary guideline for the sampling and redistribution of genetic diversity for *ex situ* conservation, rehabilitation initiatives, and commercial cultivation (*Hufford & Mazer, 2003*; *Laikre et al., 2010*; *Potts, 2017*; *Schipmann et al., 2005*).

## METHODS AND MATERIALS

### Comparing the geographic distribution of genetic lineages with respect to CoEs

To test whether there is a significant pattern for Cape floral lineages to exhibit phylogeographic breaks across the boundaries of CoEs (*Bradshaw, Colville & Linder, 2015*), all peer reviewed, published studies found using the search terms 'population structure', 'population differentiation', 'genetic structure', 'phylogeography', 'geographic divergence', 'population divergence', and 'adaptive divergence' of CFR plant taxa were compiled by initially consulting reviews of Cape phylogeography (*Lexer et al., 2013*; *Tolley et al., 2014*) followed by searching online databases using the Google scholar search engine (https://scholar.google.co.za/). A broad definition of phylogeography was adopted at this stage due to the general lack of intraspecific phylogeographic literature focusing on CFR plant taxa and any study including a phylogenetic analysis in relation to a mapped geographic distribution of the samples analysed was included (i.e. species-level phylogenetic studies). Thus, a preliminary total of 17 studies covering intra- and interspecific phylogeographic investigations were identified. Five of the studies were, however, excluded from the investigation, three due to potential data reproducibility issues of RAPD and ISSR molecular markers (*Bergh et al., 2007*; *Heelemann et al., 2013*; *Tansley & Brown, 2000*); another due to the same data set being used in multiple publications (*Segarra-Moragues & Ojeda, 2010*; *Ojeda et al., 2015*); and a third due to lack of information on the geographic location of samples (*Latimer et al., 2009*). The remaining 12 studies were used in the CoE analysis (Table 1)—these provided a total of 179 populations for the between versus within CoE population pair comparisons (raw population scoring data is available online at DOI 10.6084/m9.figshare.11370468.v1) and included a range of molecular marker types (Microsatellite, Sanger Sequencing, Next-Generation Sequencing, and AFLP markers, summarised in Table 1).

**Table 1  Summary of the results from the 12 data sets used to test for phylogeographic breaks across the boundaries between Centres of Endemism in the Cape Floristic Region, South Africa.**

| Genus | Family | No. species investigated (populations) | CFR sub-region | Min samples per population | Max samples per population | CoEs covered (boundaries) | Genetic break across CoEs (%) | Uncertain CoE member-ship (%) | Genetic homo-geneity across CoEs (%) | Genetic break within CoEs (%) | Primary seed dispersal mechanism | Primary pollen dispersal mechanism | Molecular methods used | Genome explored | Reference |
|---|---|---|---|---|---|---|---|---|---|---|---|---|---|---|---|
| Aspalathus | Fabaceae | 1 (5) | W | 3 | 5 | 2 (2) | 100 | 0 | 0 | 0 | Myrmecochory | Insect | Sanger sequencing | Chloroplast | Malgas et al. (2010) |
| Cyclopia | Fabaceae | 1 (3) | E | 24 | 30 | 1 (2) | 100 | 0 | 0 | 0 | Myrmecochory | Insect | Microsatellite | Nuclear | Niemandt et al. (2018) |
| Cyclopia | Fabaceae | 3 (22) | W & E | 6 | 24 | 5 (5) | 100 | 0 | 0 | 3 | Myrmecochory | Insect | Sanger sequencing | Chloroplast | Galuszynski & Potts (2020a, 2020b) |
| Erica | Ericaceae | 1 (21) | W | 10 | 30 | 2 (6) | 86 | 5 | 9 | 30 | Myrmecochory | Insect | Microsatellite | Nuclear | Segarra-Moragues & Ojeda (2010) |
| Erica | Ericaceae | 1 (14) | W | 1 | 2 | 4 (5) | 93 | 7 | 0 | 15 | Passive | Bird | Sanger sequencing | Nuclear and Chloroplast | Pirie et al. (2017) |
| Gladiolus | Iridaceae | 1 (15) | W | 4 | 11 | 4 (6) | 81 | 6 | 13 | 37 | Wind | Insect | Sanger sequencing and AFLP | Nuclear and Chloroplast | Rymer et al. (2010) |
| Leucospermum | Proteaceae | 1 (4) | W | 7 | 22 | 2 (2) | 100 | 0 | 0 | 12 | Serotinous | Bird | Sanger sequencing | Nuclear and Chloroplast | Johnson, He & Pauw (2014) |
| Protea | Proteaceae | 6 (30) | W & E | 20 | 20 | 7 (10) | 87 | 3 | 10 | 23 | Wind | Bird | Microsatellite | Nuclear | Prunier & Holsinger (2010) |
| Protea | Proteaceae | 1 (19) | W & E | 8 | 73 | 6 (11) | 71 | 7 | 22 | 10 | Wind | Bird | Next-Gen sequencing | Nuclear | Prunier et al. (2017) |
| Restio | Restionaceae | 1 (10) | W & E | 5 | 10 | 5 (6) | 100 | 0 | 0 | 0 | Wind | Wind | Next-Gen and Sanger sequencing | Nuclear and Chloroplast | Lexer et al. (2014) |
| Tetraria | Cyperaceae | 1 (36) | W & E | 2 | 24 | 7 (10) | 75 | 19 | 6 | 14 | Passive | Wind | Sanger sequencing | Nuclear and Chloroplast | Britton, Hedderson & Verboom (2014) |

To test whether genetic divergence occurs across the CoEs, each study's sample distribution map was overlaid with the CoEs and each population assigned to a centre using *QGIS (3.2.2)* (*Lacaze, Dudek & Picard, 2018*). In cases where populations occurring near the boundaries between CoEs were challenging to assign to either CoE, an approach that promotes a null-hypothesis (i.e. phylogeographic breaks do not occur across CoE boundaries) was adopted. These populations were marked as having an uncertain CoE membership and (unless genetically unique among all the populations in either possible CoEs) were considered to not exhibit a phylogeographic break across CoEs. This increased the tendency to detect cases of no phylogeographic break across CoE boundaries.

From the phylogenetic or phylogeographic analyses reported in each study, it was determined whether population pairs exhibited genetic divergence. Populations were assigned to a genetic group based on membership to a clade or sub-clade as reported in their respective studies. Population pairs occurring across adjacent CoE boundaries were then evaluated to determine if they belonged to the same, or different genetic groups. Each population pair was scored as either: (a) representing an phylogeographic break across the CoE boundary (hereafter referred to as a 'CoE phylogeographic break'), or (b) representing no phylogeographic break across the CoE boundary (hereafter referred to as 'inter-CoE homogeneity'). In addition, genetic divergence between population pairs occurring within CoEs was examined for each study using the same approach, with each possible population pair within a CoE scored as either exhibiting a phylogeographic break or homogeneity. The sample distributions of a subset of the studies used to explore phylogeographic breaks across CoEs are shown in Fig. 2, providing examples of populations marked as exhibiting: no phylogeographic break across CoE boundaries, uncertain CoE membership, or within CoE divergence.

Chi-squared tests with simulated *p*-value based on 10,000 replicates were used to determine whether the final population scoring deviated from random assignment with equal probability of population pairs exhibiting a CoE phylogenetic break or homogeneity. Tests were performed using the base package in *R (v3.5.1)* (*R Core Team R, 2018*).

## RESULTS

The final data set consisted of 540 population pairs compared across a total of 33 boundaries between adjacent CoEs. Eight genera were included in the analysis, with an average of 10 (SD = 9, range = 1–25) CoE boundaries compared per genus. The number of population pairs compared across adjacent CoEs ranged from one (six studies) to seven (two studies) with an average of 3 (SD = 2, range = 1–17) CoE boundaries compared per study. Sampling in five of the studies included in the CoE analysis were limited to the western CFR, while only one was limited to the eastern CFR and the remainders sampled across the western and eastern CFR.

More populations pairs (506 of 540) exhibited phylogenetic breaks across CoE boundaries that would be expected if there was equal probability of assignment ($x^2 = 886$, $p < 0.001$). Eight populations exhibited CoE uncertainty and were subsequently combined with the populations pairs that exhibited inter-CoE homogeneity, resulting in a total of 34 population pairs exhibiting inter-CoE homogeneity. The number of cases of

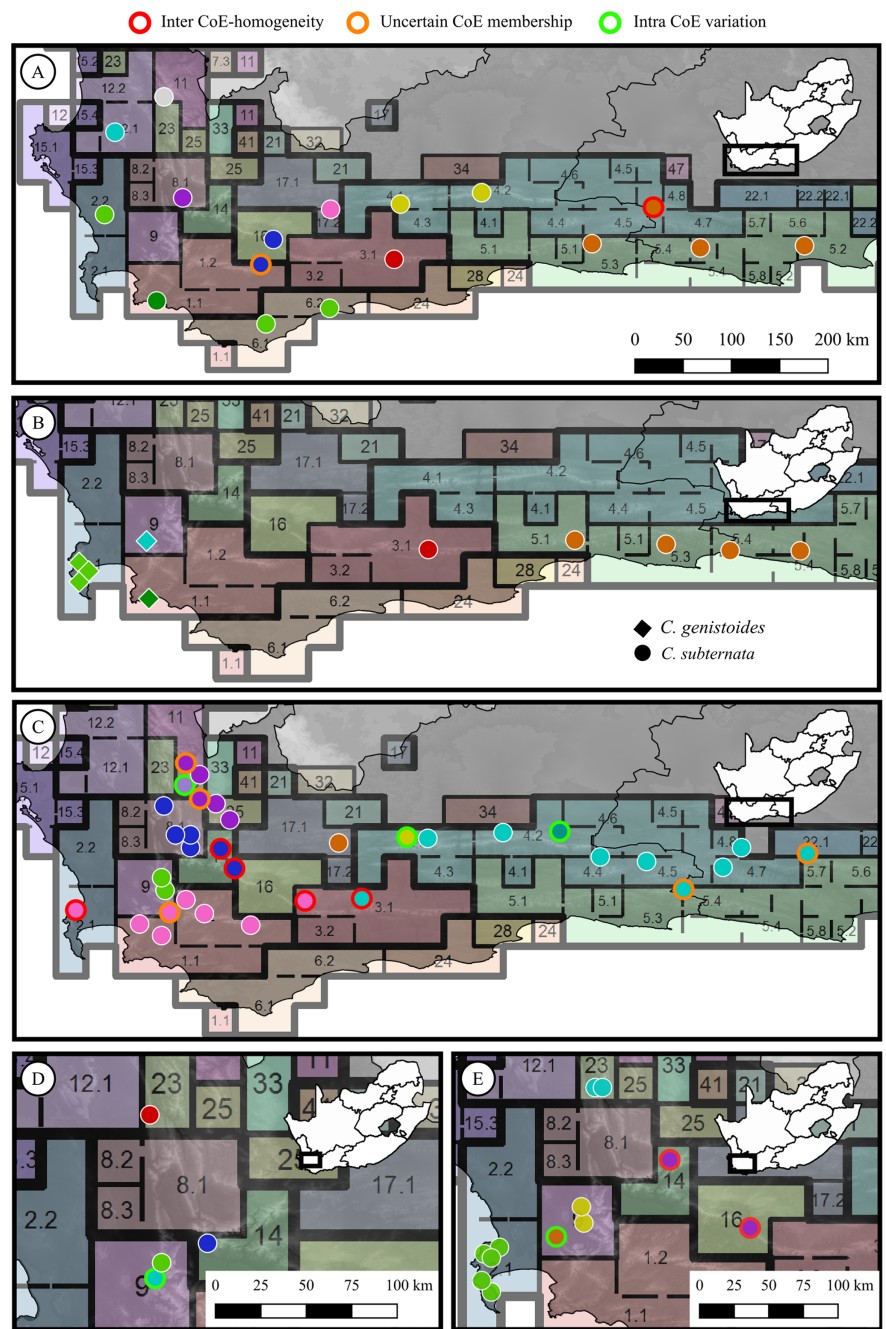

**Figure 2 Sample distributions for five of the phylogeographic studies used to test for phylogeographic breaks across Centres of Endemism.** (A) *Protea repens* (L.) L., *Prunier et al. (2017)*; (B) *Cyclopia genistoides* (L) R. Br. (diamonds) and *C. subternata Vogel* (circles), *Galuszynski & Potts (2020b)*; (C) *Tetraria triangularis* (Boeck.) C. B. Clarke, *Britton, Hedderson & Verboom (2014)*; (D) *Leucospermum tottum* (L.) R. Br., *Johnson, He & Pauw (2014)*; (E) *Erica abietina* L., *Pirie et al. (2017)*. Circle colours represent genetic groups (as determined from the original phylogeographic analysis from the population's source study), outline colour represent the population scorings used; red indicates populations that exhibit inter-CoE phylogeographic homogeneity, orange outlines indicate populations marked as having uncertain CoE membership, green outlines indicate intra-CoE genetic variation, and white outlines represent cases of inter-CoE phylogeographic breaks.

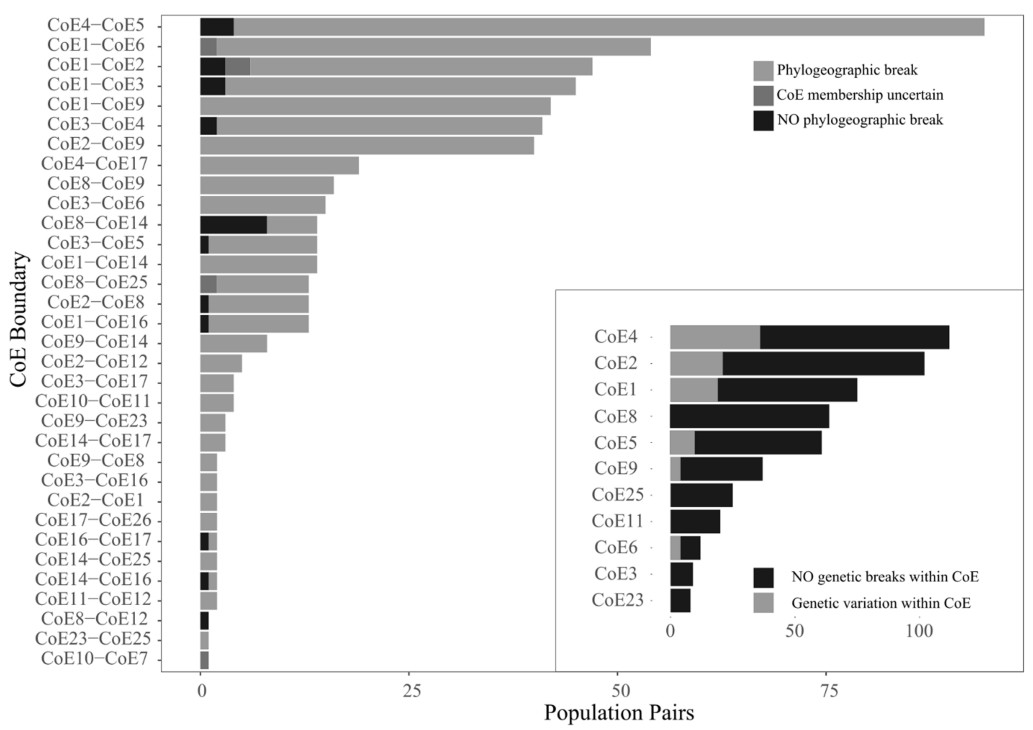

**Figure 3 Counts of the three population pair scoring options across 33 CoE boundaries.** The inset shows counts of population pairs exhibiting either intra-CoE genetic variation or not.

phylogeographic breaks, uncertainty of CoE membership and inter-CoE homogeneity for each of the CoE boundaries examined are shown in Fig. 3.

While relatively few population comparisons exhibited inter-CoE homogeneity, it is important to note that many of the cases of homogeneity involve the Hexriver Mountain Centre (CoE 14), which *Bradshaw, Colville & Linder (2015)* identified as a possible transitional zones between CoEs. The remaining cases of inter-CoE homogeneity represent isolated cases.

Within CoE divergence comparisons were conducted on a total of 619 population pairs occurring within 11 CoEs. Of these, 525 exhibited intra-CoE homogeneity (i.e. 94 pairs contained intra-CoE divergence), more than expected than if there was equal probability of assignment ($x^2 = 300$, $p < 0.001$) and shown in Fig. 3 inset.

Thus, inter-population divergence was detected in 93.7% ($n = 540$) of paired populations found on either side of a CoE boundary whereas only 15.2% ($n = 619$) were found within CoEs.

## DISCUSSION

The results of this study support the hypothesis that the evolutionary forces that have shaped the distribution of endemic plant taxa in the CFR are also operating at the intra-specific level, potentially generating shared patterns of phylogeographic structuring among plant taxa in the CFR. A significant trend was detected for population pairs to

exhibit phylogeographic breaks across CoE boundaries, while population pairs within CoEs exhibited significantly more genetic homogeneity. These results should, however, be viewed as preliminary insights into general CFR phylogeographic trends. The incomplete phylogeographic sampling across the CFR, and a coarse QDS scale mapping of the CoEs used make this study a proof of concept, which provides a novel hypothesis to be tested using more refined data sets by future workers.

Nonetheless, reducing complex species distributions to spatially distinct geographic units, or phytogeographic zones, appears to be particularly useful in species rich areas (*Edler et al., 2016*), where generating data for a representative number of individual species may be time consuming and financially restrictive. Defining phytogeographic zones based on evolutionary processes unique to the landscape under investigation, such as distribution of endemic taxa in the CFR, results in the boundaries between the defined zones possibly being representative of local evolutionary drivers (*Bradshaw, Colville & Linder, 2015*). If this is the case, as our results suggest, CoEs may provide a valuable resource for phylogeographic research, conservation planning and wild genetic resource management in the CFR. Below we briefly discuss the potential role CoEs may play in facilitating the advancement of these fields.

## The potential role of CoEs in future research

Fundamental to any successful molecular ecological study is adequate sample collection across the target species' range and within populations. Furthermore, strategic sampling of genetic variation across a species' range offers the opportunity for specific hypotheses to be statistically tested (*Morando, Avila & Sites, 2003*). However, sampling design has proven to be a challenge for phylogeographers and is often highly variable within and across studies (*Gutiérrez-García & Vázquez-Domínguez, 2011*). This is evident in the CFR, with the average minimum and maximum number of individuals sampled per population being 8 (SD = 8) and 21 (SD = 16), respectively (Table 1).

This haphazard sampling is often unavoidable in the context of the CFR, where frequent fires may rapidly reduce populations to a few individuals (*Galuszynski & Potts, 2020a*) and populations may be restricted to inaccessible mountain sites (*McDonald & Cowling, 1995*; *Schutte, Vlok & Van Wyk, 1995*). However, the realisation that the boundaries between CoEs may predict phylogeographic breaks among populations in the CFR, presents an opportunity to explore intraspecific genetic variation with targeted sampling. Sampling strategies can be planned around this information to either maximise the amount of intraspecific genetic variation detected (i.e. sample populations from different CoEs and sub-CoEs), or explore specific phylogeographic and evolutionary hypotheses (e.g. sample transects across CoE boundaries, or explore adaptive divergence of populations sampled from different CoEs).

While these suggestions act only to highlight possible applications of CoEs in molecular ecology and phylogeography in the CFR, the first step towards integrating CoEs into novel molecular research should be the remapping of CoEs at finer resolution. *Bradshaw, Colville & Linder (2015)* recognized the limitations of a coarse scale at which the

CoEs were mapped, at times resulting in the merging of lowland and mountain habitats into single CoEs and a number of poorly resolved centres. The inclusion of finer scaled distributional data in future CoE mapping may help overcome this issue of resolution and facilitate the integration of CoEs into other spatial planning and management activities in the CFR.

## Implications for conservation planning

The current regional conservation plan in the CFR (Cape Action Plan for People and the Environment, CAPE) was developed in 2003 (*Cowling et al., 2003*), prior to the publication of any of the phylogeographic studies included in this study (Table 1). Applying a systematic conservation planning approach to selecting sites of high conservation value, CAPE relied predominantly on environmental features, mapped as Broad Habitat Units (BHUs) (*Cowling & Heijnis, 2001*), and the distribution of vertebrates and Proteaceae for generating measures of irreplaceability (*Cowling et al., 2003*). In this context, irreplaceability refers to the potential contribution of a site to achieving a predetermined conservation goal, or alternatively, the extent to which achieving said conservation goal is compromised if a site is lost (*Pressey, Johnson & Wilson, 1994*). The irreplaceability of BHUs can therefore be viewed as the currency with which conservation success is ultimately measured. While the BHUs used to develop CAPE were conceived with the preservation of local evolutionary processes in mind (*Cowling & Heijnis, 2001*), environmental data was used as a proxy to represent these processes and the BHUs share little spatial congruence with the CoEs of *Bradshaw, Colville & Linder (2015)*. Consequently, current conservation targets could be failing to adequately represent intraspecific genetic diversity. Phylogeographic data should therefore be included in future conservation planning in the CFR.

The description of conservation priorities using insights gained from phylogeographic analysis has been applied to biodiverse regions elsewhere in the world. These include setting conservation priorities for the: mammal fauna of the Amazon (*Da Silva & Patton, 1998*); lizard fauna of northern Australia (*Rosauer et al., 2016*); and the floras of south western Australia (*Byrne, 2007*), California (*Calsbeek, Thompson & Richardson, 2003*), and the Mediterranean basin (*Médail & Baumel, 2018*). However, the phylogeographic studies conducted in these regions, and subsequently used to define conservation targets, were done on species with sufficient range overlap to allow for the detection of shared patterns of genetic structuring. Unfortunately, with only four phylogeographic studies conducted on widespread CFR taxa and most studies limited to species endemic to the West Coast and immediate surrounding areas (Fig. 2; Table 1), there are still areas with no available phylogeographic information (predominantly in the north werstern and eastern CFR). Thus CoE boundaries may be suitable surrogates for general phylogeographic data in aiding conservation planning until more phylogeographic research is conducted in the CFR, providing a readily available, precautionary and cost effective means of accounting for intraspecific genetic diversity in future conservation planning in the CFR.

## Spatial limits to translocation

In addition to delimiting areas for conserving extant habitat, due to the high numbers of locally endemic plant taxa (8,900 endemic species occur in the broader Cape area, *Goldblatt & Manning, 2002*), achieving conservation targets may require the reintroduction of species into previously degraded habitats (*Cowan & Anderson, 2014*; *Rebelo et al., 2011*; *Waller et al., 2015*, *2016*) or translocations into novel habitat (*Milton et al., 1999*). Furthermore, rises in consumer consciousness have increased the demand for natural products, which in turn has resulted in increased cultivation of wild crop species as a means to curb plant population decline associated with wild harvesting (*Canter, Thomas & Ernst, 2005*; *Lubbe & Verpoorte, 2011*; *Schipmann et al., 2005*). Unfortunately, the underlying levels of genetic diversity and structuring have rarely been considered during translocation activities, and may expose local populations to foreign genetic material, possibly disrupting local genetic diversity patterns (*Laikre et al., 2010*; *McKay et al., 2005*).

With no general guidelines in place to direct the redistribution of genetic material in the CFR, it is unlikely that the potential for geographically structured genetic lineages is considered during translocation activities. However, with investigations into genetic issues associated with the translocation of endemic plants only recently emerging in the literature (*Bello et al., 2018*; N.C. Galuszynski, 2018, unpublished data; *Johnson, 2018*; *Macqueen & Potts, 2018*; *Malgas et al., 2010*; *Mayonde et al., 2015*; *Potts, 2017*), it is clear that more work is required to better describe and protect wild genetic diversity in the CFR. CoEs offer a suitable proxy to develop precautionary limits to the redistribution of genetic material and design sampling strategies for describing the levels of wild genetic diversity of CFR taxa for the development of genetic resource management plans.

## CONCLUSIONS

The data sets used in this study were not developed with the intention of answering the questions posed here, and there is great potential to refine this work in future studies. Finer resolution mapping units should be used to generate CoEs. Be it with environmental layers that represent putative barriers to dispersal, modelled distributions of endemic plant taxa, which in itself includes environmental data, or more fine resolution locality data. Regardless, more refined edges between CoEs should be developed to better reflect natural boundaries between species distributions and evolutionary processes. From these refined CoEs, hypothesis-driven sampling of populations can be conducted to test for phylogeographic breaks across CoEs, facilitating explorations into the evolutionary processes that have generated the observed patterns of phylogeographic structuring and diversity in the CFR flora. Additionally, this study highlights the lack of available phylogeographic work in the CFR. Future studies should focus on exploring novel genera and widespread taxa, which will help to fill the current gaps in our phylogeographic knowledge of Cape plants.

Despite the limited data available for the current study, the overwhelming tendency for phylogeographic breaks to occur across CoE boundaries highlights an important aspect of diversity in the CFR—intraspecific genetic divergence has likely been driven by the same forces that have generated the exceptional floristic diversity of this region. CoEs do,

therefore, provide a spatial context for predicting and preserving plant phylogeographic patterns in the CFR. These findings highlight the potential value of CoEs for: developing phylogeographic research, aiding future conservation planning, and sampling wild genetic resources. Furthermore, policy should ensure that CoEs be adopted as precautionary spatial limits for the translocation of genetic lineages for rehabilitation and commercial cultivation, until data specific to the species concerned is made available.

### Funding
This work was supported by the National Research Fund of South Africa (Grant nos. 99034, 95992, 114687) and the Table Mountain Fund (Grant no. TM2499). The funders had no role in study design, data collection and analysis, decision to publish, or preparation of the manuscript.

### Grant Disclosures
The following grant information was disclosed by the authors:
National Research Fund of South Africa: 99034, 95992, 114687.
Table Mountain Fund: TM2499.

### Competing Interests
Alastair J. Potts is an Academic Editor for PeerJ.

### Author Contributions
- Nicholas C. Galuszynski performed the experiments, prepared figures and/or tables, authored or reviewed drafts of the paper, and approved the final draft.
- Alastair J. Potts conceived and designed the experiments, analysed the data, authored or reviewed drafts of the paper, and approved the final draft.

### Data Availability
Population pairwise scoring for each CoE boundary and within CoEs is available at figshare: Galuszynski, Nicholas (2020): Phylogeographic breaks within and across the boundaries of CoEs in the Cape Floristic Region. figshare. Dataset.
DOI 10.6084/m9.figshare.11370468.v1.

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
