# Peer review of "Do Centres of Endemism provide a spatial context for predicting and preserving plant phylogeographic patterns in the Cape Floristic Region, South Africa?"

_PeerJ, doi:10.7717/peerj.10045_

## Round 0.1 · original submission · Major Revisions

The hypothesis and design of the study are adequate however the published data utilized to conduct analyses is scarce. The
data set was too small to make major conclusions. The paper should emphasize that results here suggest that this field require more attention and that to reach strong conclusions more data from different taxa should be gathered and that this is an exploratory study.

Reviewer 1 ·

Basic reporting

No comment

Experimental design

No comment

Validity of the findings

No comment

Additional comments

The authors used the Centres of Endemism (CoEs) of Bradshaw et al. (2015) to test whether the geographic distribution of CFR endemic taxa predicts the distribution of intraspecific genetic variation. I commend the authors for this extensive work. The manuscript is clearly written painstakingly in professional manner. The results are compelling. The authors have shown that the evolutionary forces that have produced and maintained the exceptional plant diversity in the CFR have likely operated at the intraspecific level, producing similar patterns of phylogeographic structuring of plant lineages in spite of life history or taxonomy.

I only have the following minor observations that will hopefully help to improve the manuscript.

Line 7-9, reads “Here we test whether plant populations in the CFR exhibit phylogeographic breaks across the boundaries between Centres of Endemism (CoEs)” while line 94-96 reads “...to test whether the geographic distribution of CFR endemic taxa predicts the distribution of intraspecific genetic variation”. Although both statements may convey the same information, the use of ‘plant populations’ to refer to ‘endemic taxa’ to me sounds ambiguous since generally, not all plant populations are endemic to CFR even though the study covered only the endemic once. I suggest the use of similar terms for clarity and avoidance of ambiguity.

Line 14. Please specify the additional data used in this study. I couldn’t find it in the methodology section.

Line 22. Correct the sign of the Chi-square

Line 94 reads “Here we use 25 CFR core centres of endemism CoEs”, however, only 18 CoEs reflected in the result (see Figure 3 main). Please clarify.

Line 113. Put a space here ‘Levyn.Cowling’.

Line 130. Replace ‘that’ with ‘than’

Line 154. What does the asterisk on the “Phylogeography” mean? Also is the spelling correct?

Line 173. The reference Lacaze et al., 2018 is missing in the list of references.

Line 169. Please provide the raw data of the 179 population as supplementary file. This will ease the burden of going through all the 12 sources to find information about the species populations used in this study.

Line 220. Within CoE divergence comparisons were said to be conducted across 12 CoEs”, however, Figure 3 Inset shows 11 CoEs. Please clarify.

Reviewer 2 ·

Basic reporting

no comment

Experimental design

Please see my general comment!

Validity of the findings

no comment

Additional comments

I only have few comments/suggestions. My main concern stems from the use of very few published data, in this case only 12 studies representing 6 families and 8 genera, to study a well-designed question. The nature of the study would also need to take into consideration of different aspects of the study group regarding for example; life history traits, mating system, habitat, life forms, dispersal mechanisms and so on. I believe the authors tried their best to include some of the information in to account but not to the fullest to qualify the study as a complete one. This is in addition to the lack of well representative phylogeographic sampling across the CFR, as also stated by the authors. Nonetheless I appreciate the idea and the effort that the authors have put on to the study.

I haven’t seen a single word of hybrid or hybridization in the manuscript. Is that because none of the species group examined didn’t show any signal of hybridization or you have systematically left it out? Would be great to know how does the distribution of hybrids around CoEs? Is it common across or mainly confined within boundaries?

I was surprised not to see a single sentence explaining the type of molecular methods used to detect genetic divergence in the different studies. I suggest to include a text in the method and column on Table 1 to show the type of method used in each study.

Pp 129-130: please correct typo “more often that within” to “more often than within”

126-132: I don’t see the importance of listing the first hypothesis if you are not going to test in the in this study.

Reviewer 3 ·

Basic reporting

I am satisfied that the manuscript is clear and unambiguous and professional English was used throughout. Literature references are relevant and sufficient field background and context is provided.

The manuscript includes an introduction and background with sufficient background on centres of endemism and phylogeographic breaks in a ‘Cape’ context. Relevant prior literature was compared, evaluated and appropriately referenced. I thought the concept and ideas were explained very well.

The structure of the manuscript conforms to an acceptable format of ‘standard sections’ as required by the journal. All figures were relevant to the content of the manuscript and of sufficient resolution, and appropriately described and labelled.

The results were relevant to the hypotheses.

The manuscript sufficiently meets all the journal standards.

I though the title was misleading as it proposes to preserve phylogeographic patterns. In contrast the manuscript is highlighting a knowledge gap which involves using phylogeographic breaks to determine if genetic diversity is also influenced by steep environmental gradients at population level across SOEs.

There were few spelling and technical mistakes in the manuscript. It will require proofreading before it can be accepted for publication.

Experimental design

The research questions were well defined, relevant and meaningful. It came over very clearly how the research fills an identified knowledge gap. The approach followed is innovative by combining published research results to answer new research questions. I just think, as the authors also stated, that the data set used was too limited to really come to any meaningful conclusions.

However, the submission is very important as it clearly defines a research question, with some positive results, that requires much further research in the Cape Floristic Region. A major knowledge gap is identified, and the strength of the manuscript lies in its preliminary results and ideas on how to fill that gap for conservation purposes.

The investigation was rigorous, albeit data limited, and performed to a high technical standard. The methodological approach followed was described with sufficient detail and information to replicate.

Validity of the findings

The results are somewhat inconclusive due to the limited number of taxa considered and the low number of replicate populations. Fortunately, the authors acknowledge this shortcoming and they do not present the work as a major, definitive work on the topic, but use their positive results to encourage the scientific community to address this knowledge gap. I feel this is an important manuscript to get this idea into the scientific domain.

The data used were robust and statistically sound in terms of analyses, but not in terms of sample size. But it was sufficient to link the original research question to the results. By adjusting the title to better reflect the idea behind this manuscript, the small sample size can be justified as being part of an exploratory study.

The conclusions were appropriately stated, were connected to the original question investigated, and were limited to those supported by the results.

---

## Round 0.2 · accepted · Accept

Thanks for adding these sentences to the Abstract and in the text explaining that more phylogeographic studies are needed in the CFR to reach more supported conclusions. Also, you clarify the use of the 25 centres of endemism of Bradshaw. Both reviewers coincided that the paper should be accepted and I agree with them.

Reviewer 1 ·

Basic reporting

No comment

Experimental design

No comment

Validity of the findings

No comment

Reviewer 3 ·

Basic reporting

Corrections have been addressed appropriately.

Experimental design

Not applicable.

Validity of the findings

Corrections have been addressed appropriately.